# PET Glycolysis to BHET Efficiently Catalyzed by Stable and Recyclable Pd-Cu/γ-Al_2_O_3_

**DOI:** 10.3390/molecules29184305

**Published:** 2024-09-11

**Authors:** Lei Zhou, Enbo Qin, Hao Huang, Yuanyou Wang, Mingxin Li

**Affiliations:** 1School of Chemical Engineering, Yangzhou Polytechnic Institute, Yangzhou 225127, China; 17372976062@163.com (E.Q.); huangh0825@163.com (H.H.); 18012336591@163.com (Y.W.); 2Jiangsu Polyester Synthesis and Renewable Technology Engineering Research Center, Yangzhou 225127, China

**Keywords:** poly(ethylene terephthalate), glycolysis, monomer, molecular sieve catalyst, recycling

## Abstract

Glycolysis of poly(ethylene terephthalate) (PET) is a prospective way for degradation of PET to its monomer bis(hydroxyethyl) terephthalate (BHET), providing the possibility for a permanent loop recycling. However, most reported glycolysis catalysts are homogeneous, making the catalyst difficult to recover and contaminating the products. Herein, we reported on the Pd-Cu/γ-Al_2_O_3_ catalyst and applied it in the glycolysis of PET as catalyst. The formed structure gave Pd-Cu/γ-Al_2_O_3_ a high active surface area, which enabled these micro-particles to work more efficiently. The PET conversion and BHET yield reached 99% and 86%, respectively, in the presence of 5 wt% of Pd-Cu/γ-Al_2_O_3_ catalyst within 80 min at 160 °C. After the reaction, the catalyst can be quickly separated by filtration, so it can be easily reused without significant loss of reactivity at least five times. Therefore, the Pd-Cu/γ-Al_2_O_3_ catalyst may contribute to an economically and environmentally improved large-scale recycling of PET fiber waste.

## 1. Introduction

Poly(ethylene terephthalate) (PET) is widely used due to its attractive physical and chemical properties, including high transparency, thermal stability, mechanical properties, and a good oxygen barrier [1,2]. However, the broad commercial use of PET products is often short-term, and the accumulation of a large number of waste plastics has caused serious pollution and energy waste [3]. In addition, PET is too inert to break down by means of environmental self-purification [4]. The chemical recycling method regards waste polyester as an undeveloped chemical resource and degrades PET into a variety of high-value downstream products or raw material for regenerate PET [5]. Compared with the physical method, the chemical recycling method is more flexible and thorough, so it has more application potential [6].

Chemical degradation of PET is achieved through nucleophilic or electrophilic attacks on its carbonyl groups by different substrates, and it is usually divided into the following: alcoholysis [7,8,9], hydrolysis [10,11,12], and ammonolysis [13,14,15]. Among them, the use of alcohols as substrates, especially ethylene glycol, is the focus of current research [16,17,18]. In contrast, the PET hydrolysis process always produces a large amount of acid and alkali wastewater, and the types of products of ammonolysis are relatively narrow [19]. One of the authors has previously reported on the degradation of PET by isooctanol in the presence of zinc-based ionic liquid [9]. However, the market demand for dioctyl terephthalate (DOTP) is not large, and it cannot regenerate PET in a closed loop. To realize closed-loop recycling of PET in a sustainable way, the study of PET glycolysis to bis(2-hydroxyethyl) terephthalate (BHET) has received more attention [20,21].

It was reported that transition metal acetates have good catalytic activity for PET glycolysis, such as zinc acetate, manganese acetate, cobalt acetate, and lead acetate [17,22]. Also, ionic liquids have been widely studied for their synergistic catalytic role in PET degradation. Zhang et al. studied the effect of methyl urea/Zn(OAc)_2_ catalyst on PET glycolysis, and the reaction was performed at 170 °C for 30 min, where the BHET yield reached 82% [23]. Subsequently, they also developed metal-free choline ionic liquids such as choline/formic acid and choline/acetic acid catalytic systems, which achieved 85% BHET yield at 180 °C, but the reaction time took 4 h [8]. It is worth noting that although the homogeneous catalyst improves the reaction effect, it may be unstable under extreme conditions and is not conducive to product separation, which may also affect the quality of recycled PET. Therefore, the development of an efficient and stable PET glycolysis catalyst is of great significance for practical industrial applications [21]. Nanosized heterogenous catalysts are another type of catalyst being intensively studied. For example, Lima et al. synthesized titanate nanotubes and applied them as a catalyst for PET glycolysis [24]. Wi et al. developed silica nanoparticles doped with ZnO and CeO_2_ used in the same glycolysis process [25]. A graphene oxide–manganese oxide nanocomposite was prepared by Park et al., and the yield of BHET was obtained up to 96% [26]. The above research shows that heterogeneous catalysts have good activity and application prospects for the PET glycolysis.

Herein, we deposited Pd and Cu on γ-Al_2_O_3_ by an impregnation method to obtain the bimetal catalyst Pd-Cu/γ-Al_2_O_3_. The Pd/Cu ratio and load amount were optimized, as well as various reaction conditions. The Pd-Cu/γ-Al_2_O_3_ retained the skeleton structure of the original molecular sieve and the experimental conditions were optimized. The results showed that the catalyst demonstrates good activity for PET glycolysis, and the catalyst can be easily recycled without significant loss of activity after repeated use.

## 2. Results and Discussion

### 2.1. Characterization and Analysis of the Pd-Cu/γ-Al_2_O_3_ Catalysts

ICP analysis was conducted to determine the actual contents of Pd and Cu in the prepared samples. The results are presented in Table 1. As can be seen from Table 1, the actual Pd content in the samples ranges from 1.81% to 1.88%, deviating by 0.19% to 0.12% from the theoretical value of 2%. This discrepancy is attributed to the fact that the actual concentration of the Pd(NO_3_)_2_ solution used was lower than the theoretically calculated value, coupled with minor losses. The theoretical ratio is the expected ratio of the content of two elements when designing an experiment. Nonetheless, the Pd content remains relatively consistent among samples, ensuring comparability in their properties. For Cu in the samples, the difference between the actual and theoretical contents is minor. Under identical reaction conditions (viz. 160 °C, 30 min, 5 wt% catalyst/PET ratio, PET:EG = 1:2 molar ratio), the efficiency of various samples towards PET degradation was investigated. The results revealed that the catalytic activity increased with the increasing Cu content, achieving the optimal performance when the ratio of Pd to Cu was 1:2.5. Consequently, subsequent characterization and detailed studies were centered around Cat 4.

The Pd-Cu/γ-Al_2_O_3_ catalyst was subjected to XRD analysis, and the results are presented in Figure 1. The characteristic peaks of γ-Al_2_O_3_ at 2θ values of 37.2°, 42.6°, and 45.7° are prominently displayed, indicating the remarkable stability of the catalyst support even after the loading of Pd-Cu. The absence of distinct peaks corresponding to metallic Pd at 2θ = 40.2° and 46.8° suggests that under the current loading conditions of less than 2%, no crystalline Pd is formed [27]. This observation implies that Pd exists in a dispersed state on the surface of the support. The characteristic peak of metallic Cu emerges at 2θ = 43.0°, which is in accordance with the ICDD-PDF card No. 4-0836. Additionally, a broad peak observed between the characteristic peaks of Pd (111) and Cu (111) corresponds to the diffraction peak of a Pd-Cu alloy. This finding underscores the formation of Pd-Cu alloys under the current catalyst synthesis conditions [28]. The presence of this alloy peak indicates that Pd and Cu interact to form an alloy phase, which may have implications for the catalytic properties of the material.

Nitrogen adsorption measurements were used to study the porosities of the Pd-Cu/γ-Al_2_O_3_ catalyst, and the results are shown in Figure 2. After functionalization, the BET specific surface area of the sample was measured at 115.6 m^2^/g, which represents a slight decrease from that of pure γ-Al_2_O_3_ (134.9 m^2^/g), attributed to the occupation of some surface area by the loaded Pd and Cu. Additionally, based on the pore size distribution calculations derived from the BJH model [29], it was found that the mesoporous structure of γ-Al_2_O_3_ remained intact, with an average pore size of 6.07 nm for the catalyst.

The results of XPS surface analysis for the Pd-Cu/γ-Al_2_O_3_ catalyst are shown in Figure 3. The binding energies (BEs) were corrected for surface charging by taking the C 1s peak of contaminant carbon as a reference at 284.5 eV. According to the peak deconvolution results, the BEs of 335.8 eV and 341.5 eV correspond to the characteristic peaks of 3d_5/2_ and 3d_3/2_ for Pd0, and the BEs of 337.1 eV and 343.4 eV correspond to the characteristic peaks of 3d_5/2_ and 3d_3/2_ for Pd^2+^ [30], respectively. In addition, the Cu 2p_3/2_ peak at 932.8 eV and Cu 2p_1/2_ peak at 952.6 eV are typical for Cu0. The Cu with low surface free energy is easy to enrich on the alloy surface; thus, the presence of Cu (0) can be observed, and which is consistent with the XRD analysis. The Cu 2p_3/2_ peak at 934.5 eV and Cu 2p_1/2_ peak at 954.2 eV were attributed to Cu^2+^ [31], and the satellite peak was observed on the higher-energy side of the main Cu^2+^ peak, further corroborating the presence of Cu ions.

As shown in Figure 4, SEM and EDS mapping measurements were performed on the Pd-Cu/γ-Al_2_O_3_ catalyst. According to the SEM images, the prepared catalyst exhibits a clustered structure, comprising smooth lamellar regions intertwined with rough, porous domains, forming aggregates that contribute to enhanced structural stability while offering a larger specific surface area for improved catalytic activity. As shown in the EDS mapping of the Pd-Cu/γ-Al_2_O_3_ catalyst, besides a large amount of Al and O from γ-Al_2_O_3_ itself, a significant amount of Cu and Pd can also be observed, which provides evidence for the successful loading of the two metals. The content of Cu and Pd is 4.7 wt% and 1.8 wt%, respectively, which is basically in accord with the result of the ICP-OES.

### 2.2. Catalytic Glycolysis of PET

The effect of reaction temperature on PET glycolysis under specific conditions (viz. 5 g PET, 60 min, PET:EG = 1:4 molar ratio) using 5 wt% Pd-Cu/γ-Al_2_O_3_ catalyst (Cat 4) as the catalyst was monitored and the results are shown in Figure 5a. At a temperature of 100 °C, the PET glycolysis rate was minimal, with a BHET yield of 33.4%, but the selectivity for BHET was higher at low temperature. Upon elevating the reaction temperature from 100 °C to 180 °C, both the PET glycolysis rate and BHET yield increased, albeit with a concurrent decrease in BHET selectivity. The better C_PET_ and Y_BHET_ were obtained at 160 °C, which were 93.2% and 71.5%, respectively. When the temperature continued to rise, the CPET remained relatively modest, but the side reactions increased so that the Y_BHET_ and S_BHET_ decreased.

An investigation into the impact of reaction time on the glycolysis of PET is presented in Figure 5b, under the following reaction conditions: 5 g PET, 5 wt% catalyst/PET ratio, and PET:EG = 1:4 molar ratio at 160 °C. It can be seen that as time extended, the C_PET_ increased from 72.1% to 97.9%. The S_BHET_ and Y_BHET_ reached a maximum of 92.3% and 85.2%, respectively, when the reaction time was 80 min. After 80 min, there was a slight decrease in the S_BHET_ and Y_BHET_ due to the existing equilibrium between the BHET, dimer, and oligomer.

To investigate the effect of catalyst dosage on PET glycolysis, experiments were performed under the following reaction conditions: 5 g PET, 80 min, 160 °C, PET:EG = 1:4 molar ratio, and the results are shown in Figure 5c. They indicate that when the catalyst loading was 5 wt%, the degradation rate reached nearly the maximum value. With the continued increase in the catalyst amount, the degradation rate remained unchanged.

The effect of the EG-to-PET ratio was studied under the following reaction conditions: 5 g PET, 5 wt% catalyst, 80 min, 160 °C, and the results are shown in Figure 5d. Based on the results, a trend of initial increase followed by a decrease was observed. When EG reached five times the molar amount of PET, the maximum C_PET_ and Y_BHET_ was achieved, which was 99.2% and 86.1%, respectively. The amount of EG continued to increase and the reaction performance slightly deteriorated, which may be owing to the lower concentration of reactants and more of the side reactions.

By comparing the above diagrams, it can be found that the reaction temperature and the amount of EG have more influence on PET glycolysis, but the reaction time or catalyst amount have less influence. Considering the cost of utilization, the optimal conditions for PET glycolysis using the Pd-Cu/γ-Al_2_O_3_ catalyst are 160 °C, 80 min, 5 wt% catalyst, PET:EG = 1:5 molar ratio, with the C_PET_, S_BHET_, and Y_BHET_ being 99.2%, 86.8%, and 86.1%, respectively. The recyclability of catalysts is an important factor for the practical application. Traditional metal salt catalysts tend to dissolve in alcohols, rendering their complete separation impractical. Even trace amounts of residual catalysts can adversely affect the quality of the final products. After the reaction is completed, Pd-Cu/γ-Al_2_O_3_ can be filtered out together with the unreacted PET, and by directly adding fresh PET, the glycolysis reaction can be continued. The results of the catalyst recycling are shown in Figure 6, it was reused five times without noticeable loss of catalytic activity, and the degradation rate and yield of PET show a slight decrease but still remain at a relatively high level.

A series of characterizations were conducted on the main product of PET glycolysis. The ^1^H NMR spectrum is shown in Figure 7, and the result indicates that the chemical shifts of each hydrogen in the main product corresponded to BHET. The signals at δ 3.72 ppm and δ 4.32 ppm indicate the presence of two kinds of -CH_2_. The signal at δ 4.98 ppm represents the -OH. The signal at δ 8.12 ppm is characteristic of the protons of the benzene ring [8]. The DSC curve of the main product is depicted in Appendix A. There was a sharp endothermic peak, and the melting onset temperature was 112.5 °C, consistent with the melting point characteristics of BHET. The result of mass spectrometry is shown in Appendix A (MS (*m*/*z*): [M]+ calcd for [BHET + Na]+, 277.07; found, 277.10). It was further confirmed that the product is BHET. The yield of the product was determined by the external standard method of high-performance liquid chromatography (HPLC). A series of standard-concentration BHET solutions were prepared, and the standard curve was obtained according to the peak area of different concentrations, as shown in Appendix A. The product was confirmed by the same retention time of the HPLC as shown in Appendix A, and the concentration of the sample was calculated according to the value of its peak area into the standard curve; then, the yield was obtained.

In addition, since the by-product can be dissolved in EG, it is necessary to remove EG from the reaction liquid by rotational evaporation first, and then dissolve the residue in water to filter out the flocculent precipitation. The by-products were analyzed by gel permeation chromatography (GPC), as shown in Figure 8, and the results showed that the molecular weight ranged from 500 to 900, and half of them concentrated in 700. These residues were initially speculated to be oligomers, typically ranging from dimers to tetramers, which failed to efficiently convert into BHET upon reaching reaction equilibrium [32].

## 3. Materials and Methods

### 3.1. Materials

PET pellets (2 × 2 × 2.5 mm) were purchased from Sinopec Yizheng Chemical Fibre Co., Ltd. (Yangzhou, China). Ethylene glycol (EG), γ-Al_2_O_3_, ethanol, cupric nitrate, palladium nitrate, etc. were purchased from XFNANO (Nanjing, China), Aldrich Chemical Company (St. Louis, MO, USA), Energy Chemical Company (Shanghai, China), or Tokyo Chemical Industry Company (Tokyo, Japan). All reagents were used as received without further treatment.

### 3.2. Preparation of Pd-Cu/γ-Al_2_O_3_ Catalysts

To prepare a series of Pd-Cu/γ-Al_2_O_3_ catalysts with varying Pd to Cu ratios using the incipient wetness impregnation method, equal masses of γ-Al_2_O_3_ were separately added into solutions containing predetermined amounts of Pd(NO_3_)_2_ and varying quantities of Cu(NO_3_)_2_. The samples were heated at 80 °C under continuous stirring until complete evaporation. Subsequently, the obtained samples were ground into powder and calcined at 450 °C for 4 h. All catalysts were prepared with a uniform Pd loading of 2%. The desired Pd/Cu ratios were set at 1:1, 1:1.5, 1:2, 1:2.5, and 1:3 for the preparation of catalyst samples denoted as Cat1, Cat2, Cat3, Cat4, and Cat5, respectively. The precise metal loadings were determined by ICP analysis.

### 3.3. General Procedure for the PET Glycolysis

In each experiment, 5 g of PET pellets were degraded, using EG (ethylene glycol) in the presence of Pd-Cu/γ-Al_2_O_3_ catalysts. All depolymerization trials were performed in a 100 mL double-necked round-bottom glass reactor, which was fitted with a thermometer, a magnetic stirrer, and a reflux condenser to maintain the reaction conditions. The depolymerization reactions proceeded under atmospheric pressure across a temperature range of 100 to 180 °C, with reaction durations spanning from 30 to 120 min. After the reaction was completed, the heterogeneous catalyst and the unreacted PET were uniformly recovered during the first filtration. In the catalyst cycle test, new PET is generally added directly to continue the reaction. Larger PET fragments can also be removed with tweezers, and the remaining catalyst can be dried to obtain a clean catalyst, but this will cause a slight loss. After cooling, perform a secondary filtration on the filtrate. The filter residue is BHET, and it was dissolved in an appropriate amount of hot water, followed by cooling and crystallization. Then, it was filtered to obtain white needle-like pure BHET product. The separation yield can be calculated, and its structure can be characterized. The conversion of PET was then calculated according to the following Equation (1):(1)$$C_{PET}\;=\;Conversion\;of\;PET=\frac{W_{0}-W_{1}}{W_{0}}\times 100\%$$
where *W*_0_ represents the initial weight of PET and *W*_1_ represents the weight of undegraded PET. Moreover, the selectivity and yield of BHET are defined by Equations (2) and (3):(2)$$S_{BHET}\;=\;Selectivity\;of\;BHET=\frac{n_{BHET}}{n_{units}}\times 100\%$$
(3)$$Y_{BHET}\;=\;Yield\;of\;BHET=S_{BHET}\;\times \;C_{PET}\;\times \;100\%$$
where *n_BHET_* represents the number of moles of BHET, *n_units_* represents the number of moles of the degraded PET units, *S_BHET_* represents the selectivity of BHET, and *C_PET_* represents the conversion of BHET.

### 3.4. Characterization of Catalysts and Products

The prepared Pd-Cu/γ-Al_2_O_3_ catalysts and the PET degradation products were characterized by the inductively coupled plasma optical emission spectrometer (ICP-OES) Avio 200 (PerkinElmer, Waltham, MA, USA). In addition, 1H nuclear magnetic resonance (^1^H NMR) spectra were recorded on an AVANCE-III 600 NMR spectrometer operating at 400 MHz in DMSO-d6 solution. Differential scanning calorimetry (DSC) scans were obtained using the DSC-1 STARe system by heating from 0 °C to 200 °C at a heating rate of 5 °C min^−1^ in an atmosphere of nitrogen with a flow rate of 20 mL·min^−1^. The mass spectra were carried out on a micro-TOF instrument (Bruker, Karlsruhe, Germany) equipped with electron ionization (ESI). HPLC analysis of the main product was performed using an ACQUITY HPLC which was equipped with a refractive index detector and BET C18 column (Waters, Milford, MA, USA) under the condition of a column temperature of 25 °C, detector temperature of 40 °C, solvent methanol/water ratio (40:60), and flow rate of 0.3 mL·min^−1^. Gel permeation chromatography (GPC) analysis was performed on a PL-GPC 50 system (Agilent, Santa Clara, CA, USA) under the conditions of an oven temperature of 30 °C, solvent trichloromethane, and a flow rate of 1.0 mL·min^−1^. Three columns (PLgel 5 μm guard, length 50 mm, 7.5 mm i.d.; PLgel 5 μm MIXED-C, length 300 mm, 7.5 mm i.d.; and PLgel 5 μm MIXED-D, length 300 mm, 7.5 mm i.d.) were used. The detector used on the GPC was a refractive index (RI) detector. X-ray photoelectron spectroscopy (XPS) analysis was performed by PHI 5000 Versa Probe (ULVAC-PHI, Kanagawa, Japan). X-ray diffraction (XRD) analysis was performed by a D/max-RA X-ray diffractometer (RigaKu, Tokyo, Japan), with the following conditions: Cu Kα, λ = 1, 54,056 Å, the scan speed is 0.5–5 °·min^−1^, 40 kV, 40 mA. Scanning electron microscopy (SEM) and EDS mapping were performed by Gemini SEM 300 (Oberkochen, Germany) and Oxford X-MAX (Oxford, UK), respectively.

## 4. Conclusions

In summary, Pd and Cu were successfully loaded onto γ-Al_2_O_3_ by dipping calcination, and the optimal Pd and Cu loads were 1.88% and 4.96%, respectively. Multiple characterization showed that the supported catalyst retains the skeleton structure of the original molecular sieve, with a large specific surface area, and the two metals are uniformly supported. The Pd-Cu/γ-Al_2_O_3_ catalyst was used in the PET glycolysis process. After optimizing the experimental conditions (viz. 160 °C, 80 min, 5 wt% catalyst, PET:EG = 1:5 molar ratio), the C_PET_, S_BHET_, and Y_BHET_ can reach 99.2%, 86.8%, and 86.1%, respectively. In addition, it was confirmed that the by-product of PET glycolysis is the oligomer that did not transform due to the reaction equilibrium. In addition, the prepared catalyst is also recyclable; it can be easily recycled without significant loss of reactivity at least five times, which helps to design more practical processes for the industry than traditional homogeneous catalysts.

## Figures and Tables

**Figure 1 molecules-29-04305-f001:**
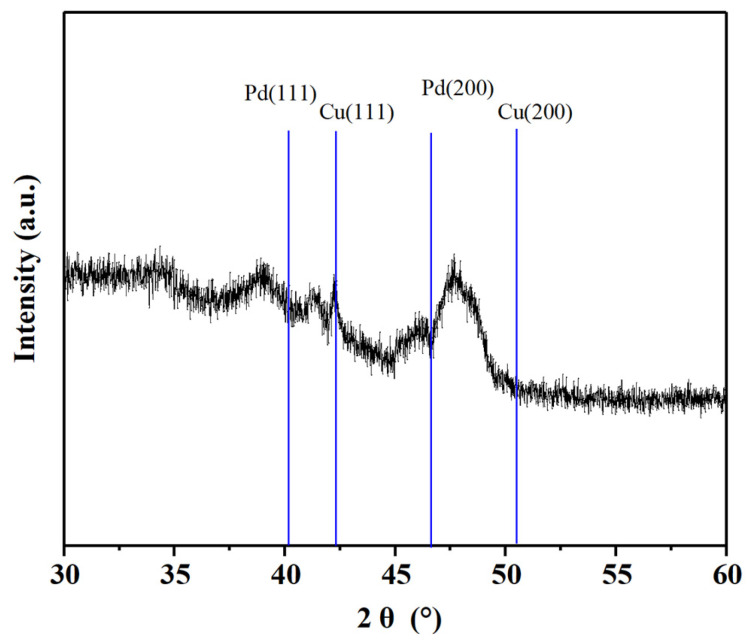
XRD pattern of Pd-Cu/γ-Al_2_O_3_ catalyst.

**Figure 2 molecules-29-04305-f002:**
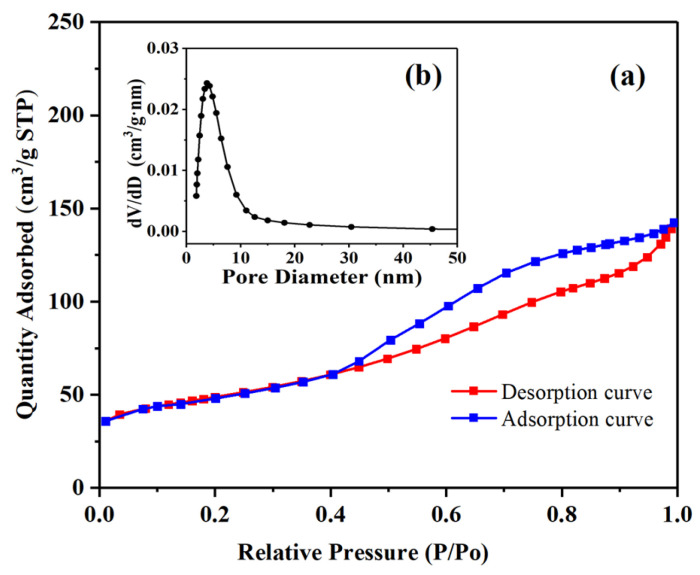
(**a**) N_2_ adsorption/desorption isotherms for Pd-Cu/γ-Al_2_O_3_ catalyst. (**b**) Pore width distribution determined by BJH method.

**Figure 3 molecules-29-04305-f003:**
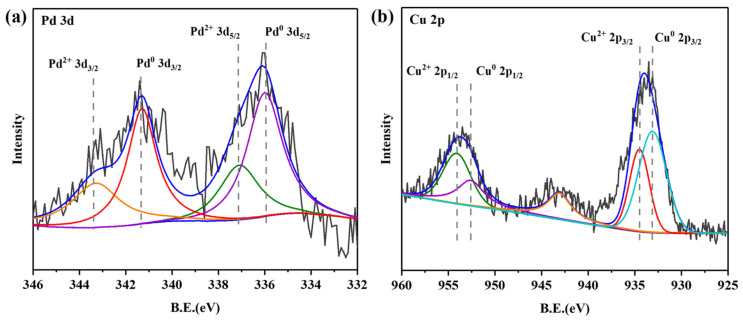
XPS spectra of Pd (**a**) and Cu (**b**) of Pd-Cu/γ-Al_2_O_3_ catalyst.

**Figure 4 molecules-29-04305-f004:**
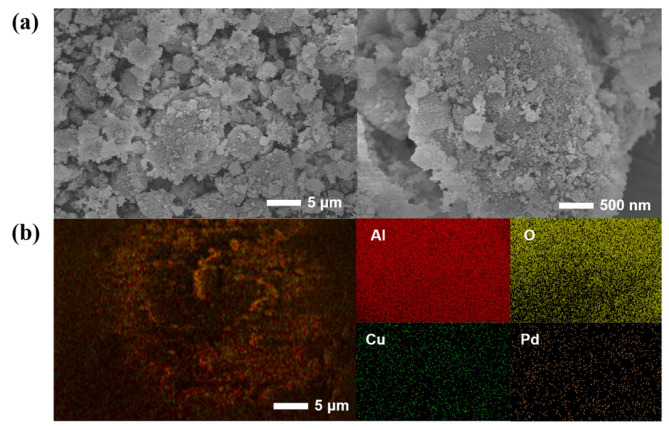
(**a**) SEM images and (**b**) EDS mapping of Pd-Cu/γ-Al_2_O_3_ catalyst.

**Figure 5 molecules-29-04305-f005:**
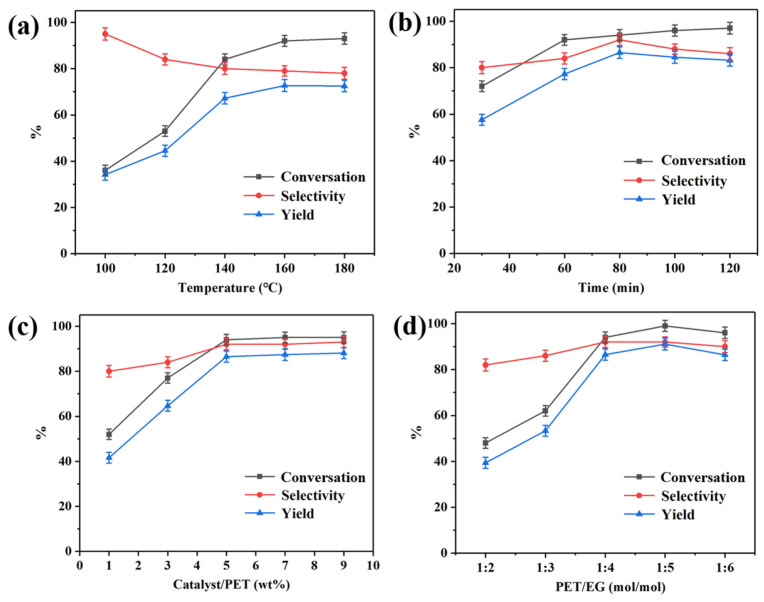
(**a**) Effect of temperature, reaction conditions: 60 min, 5 wt% catalyst, PET:EG = 1:4 molar ratio. (**b**) Effect of time, reaction conditions: 160 °C, 5 wt% catalyst, PET:EG = 1:4 molar ratio. (**c**) Effect of catalyst amount, reaction conditions: 160 °C, 80 min, PET:EG = 1:4 molar ratio. (**d**) Effect of EG amount, reaction conditions: 160 °C, 80 min, 5 wt% catalyst.

**Figure 6 molecules-29-04305-f006:**
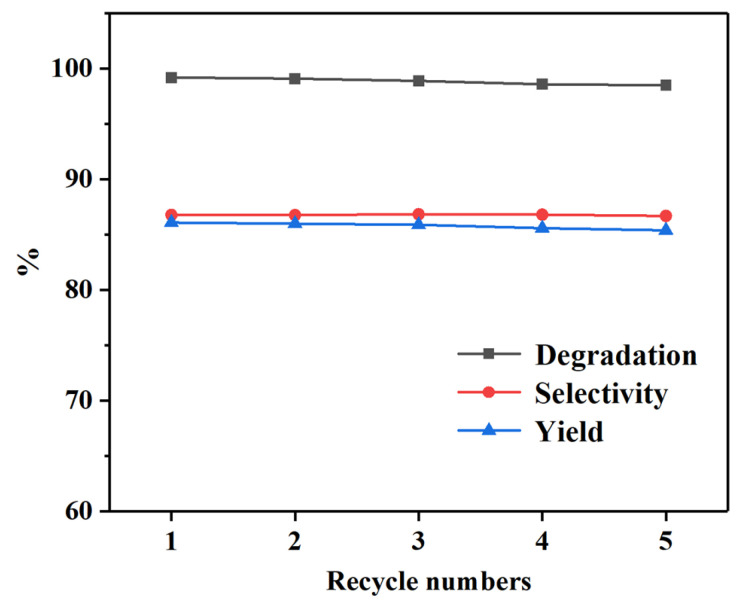
Recycling of Pd-Cu/γ-Al_2_O_3_ on the PET glycolysis. Reaction conditions: 160 °C, 80 min, 5 wt% catalyst, PET:EG = 1:5 molar ratio.

**Figure 7 molecules-29-04305-f007:**
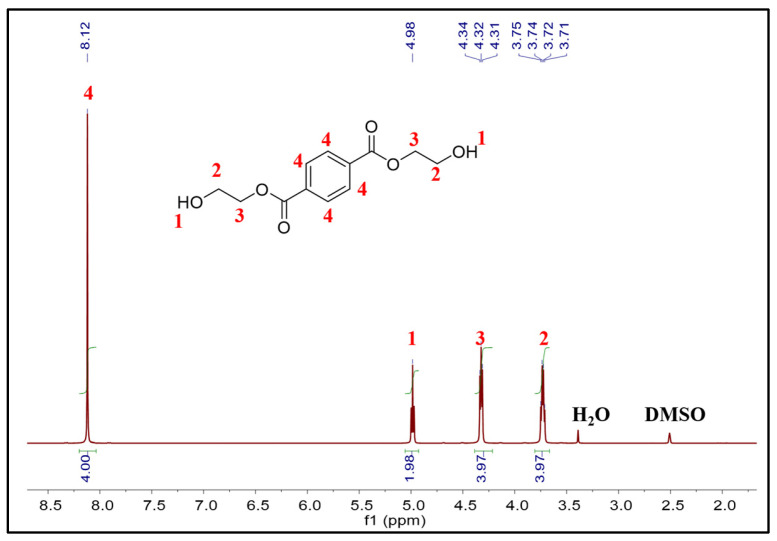
^1^H NMR spectrum of the main product of PET glycolysis in DMSO.

**Figure 8 molecules-29-04305-f008:**
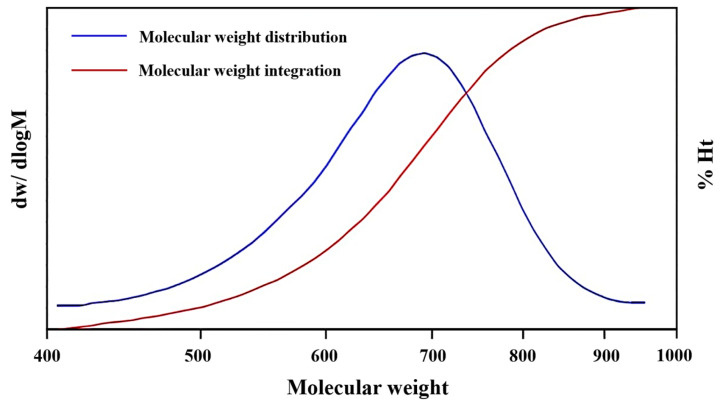
The molecular weight distribution and integration plots of by-products.

**Table 1 molecules-29-04305-t001:** ICP results of Pd and Cu elements in the Pd-Cu/γ-Al_2_O_3_ catalysts.

Entry	Cat 1	Cat 2	Cat 3	Cat 4	Cat 5
Theoretical ratio	1:1	1:1.5	1:2	1:2.5	1:3
Actual value (%)	1.85:1.98	1.81:2.96	1.88:3.95	1.88:4.96	1.87:5.94
PET degradation rate	36%	49%	57%	62%	63%

## Data Availability

The original contributions presented in the study are included in the article/Appendix A, further inquiries can be directed to the corresponding authors.

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
