# Peer review of "PET Glycolysis to BHET Efficiently Catalyzed by Stable and Recyclable Pd-Cu/γ-Al2O3"

_molecules, 2024, doi:10.3390/molecules29184305_

Round 1

Reviewer 1 Report

Comments and Suggestions for Authors

The given manuscript presents the Pd-Cu/γ-Al2O3 heterogeneous catalyst for recovery of bis(hydroxyethyl)terephtalate by depolymerisation of poly(ethylene terephthalate). Heterogeneous catalyst offers several advantages over homogeneous catalyst in terms of catalyst recovery and purity of the product obtained. The authors showed that the Pd-Cu/γ-Al2O3 catalyst is active and forms selectively the desired product. In this regard, the topic is of current importance and the presented work is interesting for the readers of Molecules.

However, some improvements should made bevor the work can be accepted for publication. Especially, the authors should pay more attention to grammar.

Here are some specific comments on improving the manuscript:

1.      The extended information about further heterogeneous catalyst used in PET glycolysis would improve the quality of the work.

2.      page 1, line 29-30: I do not understand which role the waste polyester plays as an undeveloped chemical resource in degradation of PET?

3.      Page 2, line 73-74: the deviation from theoretical value of 2% by 0.06% and 0.10% is not correct (1.81% from 2% and 1.88% from 2%). Please check!

4.      table 1: the description to table 1 is not complete. What does the information about the theoretical ratio refer to?

5.      Page 2, line 92-94: I am not sure that the presence of one peak between the characteristic peaks of Pd and Cu is sufficient evidence for the formation of an PdCu alloy. No further evidences, e.g. in XPS spectra, were found?

6.      Figure 4a: it seems to me that two SEM images were made on two different materials. The morphology seems to be different. Please check.

7.      Page 6, line 187-189: usually all results showed in Supplementary Information (figures, tables) are mentioned or shortly discussed in the main text.

8.      Page 8, line 219: the abbreviation “EC” was not described.

9.      Page 8, line 235: consider units of SBHET (%), CPET (%) and Yield (%)?

Comments on the Quality of English Language

Here are some specific comments on grammatical and syntax mistakes (the list is not complete):

1.      Page 3, line 101-104: improve the grammar!

2.      Page 3, line 104-105: check the grammar.

3.      Page 5, line 133: improve the grammar.

4.      Page 5, line 143-145: improve the grammar.

5.      Figure 5: the term “degradation” was not defined. Use the term “conversation”.

6.      The author always used “shown as Figure” which is linguistically not fully correct. Better is “shown in Figure”.

7.      Page 7, line 196-197: improve the grammar.

8.      Page 8, line 224-228: change the style of the synthesis description. E.g., not “dissolve it”, but “the residue should be dissolved”.

9.      Page 8, line 234, 235: define SBHET and CPET.

1 Page 8, line 251: consider correct unit “mL min-1”.

1  Page 9, line 270: you proved that the by-product “is the oligomer”, do not use “should be oligomer”.

Reviewer 2 Report

Comments and Suggestions for Authors

The work is a study in the field of polyethylene terephthalatechemical recycling catalysis. The topic of the study is undoubtedly relevant. References contain a sufficient number of articles for the last 5 years. In general, the content of the work does not cause any complaints. I would like to see in the text of the work a larger commentary regarding the separation of the catalyst from the reaction mixture by filtration, mentioned in the abstract. I also recommend that the authors check the spaces between symbols, in particular, °C (51), = (268), and so on.
